# Anti-Obesogenic Effects of Sulforaphane-Rich Broccoli (*Brassica oleracea var. italica*) Sprouts and Myrosinase-Rich Mustard (*Sinapis alba* L.) Seeds In Vitro and In Vivo

**DOI:** 10.3390/nu14183814

**Published:** 2022-09-15

**Authors:** Xiao Men, Xionggao Han, Se-Jeong Lee, Geon Oh, Keun-Tae Park, Jong-Kwon Han, Sun-Il Choi, Ok-Hwan Lee

**Affiliations:** 1Department of Food Biotechnology and Environmental Science, Kangwon National University, Chuncheon 24341, Korea; 2Research and Development Center, Milae Bioresourece Co., Ltd., Seoul 05542, Korea

**Keywords:** sulforaphane, bisphenol A, adipogenesis, lipolysis, anti-obesogenic

## Abstract

Glucoraphanin (GRA), a glucosinolate particularly abundant in broccoli (*Brassica oleracea var. italica*) sprouts, can be converted to sulforaphane (SFN) by the enzyme myrosinase. Herein, we investigated the anti-obesogenic effects of broccoli sprout powder (BSP), mustard (*Sinapis alba* L.) seed powder (MSP), and sulforaphane-rich MSP-BSP mixture powder (MBP) in bisphenol A (BPA)-induced 3T3-L1 cells and obese C57BL/6J mice. In vitro experiments showed that MBP, BSP, and MSP have no cytotoxic effects. Moreover, MBP and BSP inhibited the lipid accumulation in BPA-induced 3T3-L1 cells. In BPA-induced obese mice, BSP and MBP treatment inhibited body weight gain and ameliorated dyslipidemia. Furthermore, our results showed that BSP and MBP could activate AMPK, which increases ACC phosphorylation, accompanied by the upregulation of lipolysis-associated proteins (UCP-1 and CPT-1) and downregulation of adipogenesis-related proteins (C/EBP-α, FAS, aP2, PPAR-γ, and SREBP-1c), both in vitro and in vivo. Interestingly, MBP exerted a greater anti-obesogenic effect than BSP. Taken together, these findings indicate that BSP and MBP could inhibit BPA-induced adipocyte differentiation and adipogenesis by increasing the expression of the proteins related to lipid metabolism and lipolysis, effectively treating BPA-induced obesity. Thus, BSP and MBP can be developed as effective anti-obesogenic drugs.

## 1. Introduction

Obesity, a chronic metabolic disease, is a global health problem that is influenced by the interaction between environmental and hereditary factors [1]. Obesity can lead to disturbances in energy metabolism, resulting in excessive accumulation of fat, mainly due to the hypertrophy and hyperplasia of adipocytes [2]. Obesity greatly increases the risk of cancer, hypertension, cardiovascular and cerebrovascular diseases, diabetes, and osteoarthritis [3]. The most common causes of obesity include high energy diets, overeating, and a poor lifestyle, such as physical inactivity [4,5]. However, the continued increase in the global obesity burden cannot be explained solely on the basis of excessive food intake and reduced physical activity [6]. Studies showed that few individuals are unable to lose or maintain weight, regardless of an effective diet and/or increased physical activity [7]. Recent studies have found that endocrine-disrupting chemicals (EDCs) might be a significant factor in obesity. EDCs are hormone analogs present in the environment, and their molecular structure is very similar to that of hormones. EDCs can mimic hormones and interfere with the endocrine system of the body to promote the adipocyte differentiation and lipid accumulation in adipocytes that ultimately leads to obesity [8,9]. In addition, EDCs can also change the energy balance and basal metabolism by regulating the lipid metabolism. EDCs indirectly act on adipocytes by changing the way hormones control appetite, thereby promoting obesity [10,11,12]. Therefore, EDCs are also called obesogens. Accumulating evidence from in vitro and in vivo experiments, epidemiological, and human studies suggests that a variety of EDCs are potential obesogenic factors, which are closely related to obesity development. These obesogens include polychlorinated biphenyls, bisphenols, organotins, phthalates, tributyltin, some phytoestrogens, pesticides, etc. [13,14,15]. Representative obesogens in our food include organotins, phthalates, and bisphenols [13,16]. Bisphenol A (BPA), in particular, is highly associated with obesity [17,18] and can act as an active agonist of the glucocorticoid or estrogen receptors to promote adipogenesis [9,19].

Sulforaphane (SFN), an isothiocyanate present in cruciferous plants, is abundant in broccoli (*Brassica oleracea var. italica*) sprouts and produced by the hydrolysis of glucoraphanin (GRA), through the action of enzyme myrosinase [20,21]. SFN possesses pharmacological properties, including anti-cancer, antioxidant, anti-inflammatory, anti-hypertension, cardioprotective, and neuroprotective properties [22,23,24,25]. Martins et al. reported that SFN could inhibit adipocyte differentiation, promote lipolysis, and decrease triglyceride accumulation in adipocytes [26]. Moreover, SFN could prevent and treat obesity by stimulating oxidative utilization and glucose uptake, upregulating fatty acid oxidation pathways, and increasing lipolysis, while inhibiting triglyceride synthesis pathways [27]. Furthermore, SFN induces apoptosis in the adipocytes in a concentration-dependent manner [28]. However, there are only a few studies on the anti-obesogenic activity of SFN in BPA-induced obese C57BL/6J mice and 3T3-L1 cells. Although SFN can be synthesized by chemical methods, some highly toxic substances are needed in the process, and the final products of these reactions needs to be further purified, which makes their utilization difficult in foods [29]. Therefore, naturally produced SFN is a better alternative. Naturally occurring SFN is either produced endogenously or exogenously by hydrolysis. Endogenous production of SFN is known to be less effective, and the reaction is susceptible to temperature, pH, and epithiospecifier proteins to form SFN nitriles [30,31]. However, exogenous hydrolysis can overcome these drawbacks. Myrosinase and GRA can be individually extracted from their respective sources, while SFN can be obtained through a tightly controlled reaction [30].

Herein, we investigated the anti-obesogenic effects of broccoli sprouts powder (BSP) with a high GRA content, mustard (*Sinapis alba* L.) seed powder (MSP) that has a high myrosinase activity, and sulforaphane-rich MSP-BSP mixture powder (MBP) in vitro and in vivo. The anti-obesogenic effects of MBP, BSP, and MSP in BPA-induced 3T3-L1 cells and obese mice were determined by analyzing the body weight, lipid profile, lipid accumulation, and protein expression.

## 2. Materials and Methods

### 2.1. Chemicals

Insulin, N-acetyl-L-cysteine (NAC), dexamethasone (DEX), SFN standards, oil red O (ORO), BPA, and 3-isobutyl-1-methylxanthine (IBMX) were purchased from Sigma-Aldrich Co. (St. Louis, MO, USA). Normal diet (ND) and corn oil were purchased from a local grocery market (Chuncheon, Korea). Trypsin-ethylenediaminetetraacetic acid (EDTA), fetal bovine serum (FBS), bovine serum (BS), phosphate-buffered saline (PBS), Dulbecco’s modified Eagle’s medium (DMEM), and penicillin-streptomycin (P/S) were obtained from Gibco (Grand Is-land, NY, USA).

### 2.2. Preparation of MBP, BSP, and MSP

As described in our previous study [32], the same experimental method was used in this study to obtain MSP, BSP, and MBP experimental samples, and SFN and GRA contents in MBP, BSP, and MSP were analyzed by HPLC. GRA and SFN were not detected in MSP. The SFN content in MBP was 162.29 ± 1.24 μmol/g, and GRA was not detected. GRA content in BSP was 131.11 ± 1.84 μmol/g, and SFN was not detected.

### 2.3. Cell Culture

In our previous study, 1 μM DEX, 1 μg/mL insulin, and 0.5 mM IBMX (MDI) were used to induce adipocyte differentiation [32]. BPA is one of the endocrine disrupting chemicals that mimics the bioidentical hormones and acts as an active agonist of glucocorticoid receptors to promote adipogenesis [9,19]. DEX is a synthetic glucocorticoid that can induce the expression of adipocyte differentiation-related factors (PPAR-γ, CEBP-α) to promote adipocyte differentiation and lipid accumulation [33]. Therefore, in vitro experiments, in the absence of DEX, BPA was selected as an inducer of adipocyte differentiation to induced adipocytes differentiation and promote fat accumulation. Additionally, the effects of MBP, BSP, and MSP on adipogenesis and fat droplet accumulation during BPA induced 3T3-L1 preadipocyte differentiation.

Cells were cultured in DMEM containing 1% P/S and 10% BS in an incubator containing 5% CO2 and 37 °C. Cells were exposed to medium containing 20 μM BPA 6 days before starting differentiation; on the second day, after reaching confluence (day 0), cell differentiations were induced with 20 μM BPA and 1.0 μg/mL insulin, and 0.5 mM IBMX (MI) in DMEM containing 1% P/S and 10% FBS. After 48 h, the medium was changed to DMEM containing 1% P/S, 20 μM BPA, 1.0 μg/mL insulin, and 10% FBS to induce further differentiation. The medium was replaced every two days until day 10. To evaluate the anti-obesogenic effects of MBP, BSP, and MSP, cells were treated with MI and 20 μM BPA, along with MSP, BSP, or MBP, in culture medium on day 0 to induce differentiation. After 10 days of treatment, 3T3-L1 adipocytes were harvested for subsequent experiments.

### 2.4. Cell Viability Assay

Cell viability was determined by 2,3-bis(2-methoxy-4-nitro-5-sulfophenyl)-2H-tetrazolium-5-carboxanilide (XTT) assay (WelGene, Seoul, Korea). Briefly, 3T3-L1 preadipocytes were treated with different concentrations of MBP, BSP, and MSP for 10 days. A total of 1 mL of XTT and 20 μL N-methyl dibenzopyrazine methyl sulfate (PMS) was mixed and added to a 96-well plate for 4 h at 37 °C. Absorbance was measured at 450 and 690 nm.

### 2.5. Oil Red O Staining Assay

ORO was used to quantify lipid accumulation in adipocytes. Briefly, after 10 days, the medium was aspirated from 24-well plates, and the cells were washed twice with PBS. Cells were fixed for 60 min by slowly adding 500 μL of 10% formaldehyde solution to each well. Fixed cells were then washed with 60% isopropanol and dried completely at room temperature. The dried cells were stained with ORO solution for 30 min, washed with DW, and dried at room temperature. The red-stained lipids were eluted with 100% isopropanol, and the absorbance was measured at 490 nm.

### 2.6. Western Blot Analysis

Epididymal adipose tissue and cells were lysed using the protein lysis buffer and centrifuged at 10,000× *g* at 4 °C for 20 min. Protein concentration was determined using the Bio-Rad protein assay kit. Samples were then subjected to 10% sodium dodecyl sulfate-polyacrylamide gel electrophoresis (SDS-PAGE) and transferred to a polyvinylidene difluoride membrane at 100 V for 90 min. The membranes were blocked with 5% skim milk for 1 h. After blocking, the membranes were washed three times with 1× TBST and incubated with primary antibody (anti-fatty acid synthase (FAS), anti-carnitine pal-mitoyltransferase-1 (CPT-1), anti-acetyl-CoA carboxylase (ACC), anti-CCAAT/enhancer-binding protein-α (C/EBP-α), anti-sterol regulatory element-binding protein-1 (SREBP-1), anti-phospho-acetyl-CoA carboxylase (p-ACC), anti-β-actin, anti-AMP-activated protein kinase (AMPK), anti-uncoupling protein-1 (UCP-1), anti-adipocyte protein 2 (aP2), anti-peroxisome proliferator-activated receptor-γ (PPAR-γ), and anti-p-AMPK (Cell Signaling Technology; Danvers, MA, USA) (1:1000 dilution)) at 4 °C overnight. After incubation with the primary antibody, the membrane was incubated with secondary antibody from rabbit or mouse (1:2000 dilution) at room temperature for 1 h. Then, the membrane was gently washed thrice with 1× TBST. After development with ECL reagents, the images of protein bands were detected using ChemiDoc imaging system (Bio-Rad Laboratories, Inc., Hercules, CA, USA).

### 2.7. Animals

This animal research protocol has been reviewed and approved by the Institutional Animal Care and Use Committee (IACUC) of Kangwon National University. The C57BL/6J mice were maintained in accordance with Kangwon National University Regulations on Animal Experimentation (IACUC approval number: KW-210112-2). Four-week-old male C57BL/6J mice, weighing approximately 20 g, were purchased from DBL Co., Ltd. (Eumseong, Korea). Mice were housed in experimental rooms under a 12–12 h dark–light cycle, 24 ± 5 °C temperature, and 55 ± 5% relative humidity, as well as acclimated for 1 week before the experiment. All mice were provided free access to water and food (10% kcal fat) for 12 weeks.

### 2.8. Experimental Design

Mice were randomly divided into six groups (*n* = 5), as follows: normal diet (ND), BPA (BPA, 500 μg/kg/day), BPA administered with Garcinia *cambogia* extract (Gar) (BPA + Gar; Gar, 100 mg/kg/day), BPA administered with MSP (BPA + MSP; MSP, 15 mg/kg/day), BPA administered with BSP (BPA + BSP; BSP, 150 mg/kg/day), and BPA administered with MBP (BPA + MBP; MBP, 100 mg/kg/day) groups. Samples (Gar, MSP, BSP, and MSP) were dissolved in DW, and BPA was dissolved in corn oil and orally administered for 12 weeks. Body weight was measured every seven days. After 12 weeks, the mice were anesthetized with isoflurane. The epididymal adipose tissue, kidneys, liver, and spleen were removed, and blood was collected for further analysis.

### 2.9. Biochemical Analysis

Blood from each mouse was collected and centrifuged at 12,000× *g* at 4 °C for 20 min to obtain serum. Low-density lipoprotein cholesterol (LDL-c), total cholesterol (TC), high-density lipoprotein cholesterol (HDL-c), and triglyceride (TG) levels in the serum were determined using the clinical chemistry analyzer.

### 2.10. Histological Analysis

The dissected mouse tissues were washed with normal saline, wiped dry, and weighed. After routine processing of the epididymal adipose tissue, it was embedded in paraffin, sectioned into 3 μm thick sections, stained with hematoxylin and eosin (H&E), and analyzed by light microscopy (Olympus BX41, Tokyo, Japan) to observe the stained adipose tissue.

### 2.11. Statistical Analysis

Results were expressed as mean ± standard deviation (SD). All statistical analyses were performed using the SPSS software (version 24.0; SPSS Inc., Chicago, IL, USA). Statistical significance was analyzed using the Duncan’s multiple range test. A *p*-value < 0.05 was considered statistically significant.

## 3. Results

### 3.1. Effect of MBP, BSP, and MSP on Cell Viability and Lipid Accumulation in BPA-Induced 3T3-L1 Cells

Results of the XTT assay showed that SFN, MSP, BSP, MBP, and BPA have no cytotoxic effects on 3T3-L1 cells (Figure 1A). Since SFN, MSP, BSP, MBP, and BPA were non-toxic at all concentrations, the lowest/highest concentration was selected for further in vitro experiments. To determine the effects of MBP, BSP, and MSP on adipogenesis and fat droplet accumulation during BPA-induced 3T3-L1 pre-adipocyte differentiation, ORO staining was performed. In this study, the NAC experimental group was used as a positive control group, since NAC has a significant inhibitory effect on lipid accumulation [34]. On day 10 post-induction, cells in the MDI control group were highly stained with ORO, indicating lipid accumulation. In the MI (1.0 μg/mL insulin, 0.5 mM IBMX, without DEX) control group, the lipid accumulation in cells was reduced to 22%, compared to the MDI group. When the cells were treated with 20 µM BPA during adipogenic differentiation, the lipid accumulation increased significantly (55%), compared to the MI control group, indicating that BPA promotes lipid accumulation. In contrast, when the BPA-induced cells were treated with BSP (50 and 150 μg/mL) and MBP (25, 50, and 100 μg/mL) during the differentiation process, the lipid accumulation significantly decreased in mature cells by 47%, 46%, 40%, 33%, and 31%, respectively. However, there were no significant differences between the BPA and MSP (5 and 15 μg/mL)-treated cells (*p* > 0.05) (Figure 1B). These results suggest that MBP and BSP can effectively inhibit lipid accumulation in BPA-induced 3T3-L1 adipocytes, with MBP having a more pronounced effect.

### 3.2. MBP, BSP, and MSP Reduce the Expression of Adipogenic-Related Proteins in BPA-Induced 3T3-L1 Cells

Next, we investigated the effects of MBP, BSP, and MSP on the key proteins involved in lipid metabolism in BPA-induced 3T3-L1 cells. As shown in Figure 2A–C, the C/EBP-α, aP2, and PPAR-γ protein levels were significantly increased in BPA-treated cells, compared to the MI control group. Moreover, significant reductions in the C/EBP-α, aP2, and PPAR-γ protein expressions were observed when the BPA-induced cells were treated with BSP and MBP. Furthermore, BPA treatment suppressed that of CPT-1 and UCP-1 proteins and increased the expression of FAS protein. FAS is a key enzyme involved in lipogenesis and fatty acid synthesis, catalyzing the formation of long-chain fatty acids from malonyl-CoA and acetyl-CoA [35]. CPT-1 is a rate-limiting enzyme in fatty acid oxidation, and UCP-1 plays a key role in the regulation of thermogenesis. The upregulation of the UCP-1 and CPT-1 proteins promotes fat oxidation and energy expenditure [36]. We observed that treatment with BSP and MBP reduced the expression of FAS and increased the expression of UCP-1 and CPT-1 in BPA-induced 3T3-L1 cells (Figure 2D–F). No significant differences in the expression of these proteins were observed between MSP and BPA-treated cells (*p* > 0.05). Taken together, these results suggest that MBP and BSP negatively regulate BPA-induced pre-adipocyte differentiation by promoting the expression of lipolytic proteins and suppressing the expression of adipogenic proteins.

### 3.3. Effects of MBP, BSP, and MSP on AMPK Phosphorylation in BPA-Induced 3T3-L1 Cells

AMPK is involved in regulating adipogenesis, and its activation contributes to the suppression of adipogenesis and lipogenesis [37]. To further elucidate the mechanism underlying inhibitory effects of MBP, BSP, and MSP on adipogenesis in BPA-induced 3T3-L1 cells, we investigated the phosphorylation status of AMPK. The results showed that the p-AMPK/AMPK ratio was significantly reduced in the BPA group, compared to the MI control group (Figure 3A). This suggests that BPA inhibits AMPK phosphorylation, thereby promoting adipogenesis in 3T3-L1 cells. As shown in Figure 3B, the protein expression of p-ACC, a downstream signaling molecule of AMPK, was also significantly inhibited upon BPA treatment. Moreover, both BSP and MBP significantly increased the p-ACC/ACC ratio and enhanced AMPK phosphorylation in the BPA-induced 3T3-L1 cells. The effect was more pronounced in the MBP-treated cells, and the phosphorylated ACC levels in the MBP treatment group was comparable to that of the NAC treatment group. There were no significant differences in the levels of p-ACC and p-AMPK between the MSP- and BPA-treatment groups (*p* > 0.05). In summary, these results suggest that BSP and MBP increase p-ACC and p-AMPK levels to inhibit adipogenesis and lipogenesis in BPA-induced 3T3-L1 cells.

### 3.4. Effects of MBP, BSP, and MSP on Adipose Tissue Structure, Tissue Weight, and Body Weight in C57BL/6J Mice with BPA-Induced Obesity

To investigate the anti-obesogenic effects of MSP, BSP, and MBP, mice were administered BPA (500 μg/kg/day) or BPA supplemented with 100 mg/kg/day Gar (BPA + Gar), 15 mg/kg/day MSP (BPA + MSP), 150 mg/kg/day BSP (BPA + BSP), or 100 mg/kg/day MBP (BPA + MBP) for 12 weeks, and body weight was recorded weekly (Figure 4A). The BPA + Gar group served as the positive control group, since studies showed that Garcinia cambogia extract induces weight loss [38,39]. As expected, BPA administration successfully induced obesity in C57BL/6J mice. Mice in the BPA group showed a significantly high body weight and epididymal adipose tissue weight, compared to the ND group mice. However, GRA, BSP, and MBP treatment significantly decreased the body weight and epididymal adipose tissue weight in BPA-induced obese mice (Figure 4A,B,D). There were no significant differences in food intake and the weight of other tissues between the experimental groups (Figure 4C,E). In addition, histological analysis showed significantly larger adipocytes in the BPA group, and BPA-induced changes in adipocyte size were significantly reversed by Gar, BSP, and MBP (Figure 4F). MSP treatment had no significant effects on BPA-induced obese C57BL/6J mice (*p* > 0.05). Taken together, our results clearly demonstrate that MBP and BSP exert strong anti-obesogenic effects in BPA-induced obese mice.

### 3.5. Effects of MBP, BSP, and MSP on Serum Biochemical Indexes in C57BL/6J Mice with BPA-Induced Obesity

To further investigate the anti-obesogenic effects of MBP, BSP, and MSP, we evaluated the serum levels of HDL-c, LDL-c, TG, and TC in different mice groups. The TG levels in the BPA group were significantly higher than those in the ND group, as shown in Table 1. However, the BSP + BPA and MBP + BPA groups showed significantly lower TG levels, compared to the BPA-treated group. Furthermore, BPA treatment significantly reduced the HDL-c levels in mice, and BSP and MBP administration significantly increased the HDL-c levels in BPA-exposed mice. These results suggest that MBP and BSP treatment improves serum HDL-c and TG levels in C57BL/6J mice with BPA-induced obesity.

### 3.6. Effects of MBP, BSP, and MSP on Lipid Metabolism-Related Proteins in BPA-Induced Obese Mice

As shown in Figure 5A–E, the expression of key proteins involved in lipid synthesis, including SREBP-1C, aP2, C/EBP-α, PPAR-γ, and FAS, was significantly upregulated in the epididymal adipose tissue of the BPA-induced obese mice, whereas BSP and MBP administration significantly attenuated these changes, with MBP having a more pronounced effect, comparable to that of Gar. Moreover, a significant increase in UCP-1 and CPT-1 protein expression was also observed in BSP and MBP experimental groups, compared with BPA-induced mice (Figure 5F,G). There were no significant differences in the expression of any protein between the BPA and MSP groups (*p* > 0.05). Taken together, these data clearly demonstrate that BSP and MBP are effective in preventing BPA-induced obesity by controlling adipogenesis, lipogenesis, and fatty acid oxidation.

### 3.7. Effects of MBP, BSP, and MSP on AMPK Phosphorylation in C57BL/6J Mice with BPA-Induced Obesity

The p-AMPK/AMPK ratio in epididymal adipose tissue of the BPA-induced mice was significantly lower than that in the ND group mice. However, BSP and MBP administration significantly increased the p-AMPK/AMPK ratio in BPA-induced mice (Figure 6A,B). Similarly, the p-ACC/ACC ratio was also higher in the BSP and MBP administered mice, compared to that in BPA administered mice. There were significant differences in the expression of p-ACC/ACC and p-AMPK/AMPK proteins between MSP and BPA groups (*p* > 0.05). In summary, these results suggest that BSP and MBP exert metabolic effects in adipose tissue via the AMPK pathway.

## 4. Discussion

Several studies have confirmed that exposure to BPA and other EDCs can affect the function and development of adipose tissue and cause obesity. It should be noted that overweight status and obesity are not only due to excessive food intake and insufficient exercise, but they are also influenced by exposure to BPA and other EDCs [7,8]. Obesity is primarily due to increased adipocyte size and proliferation, factors associated with pre-adipocyte differentiation and lipogenesis, respectively [2,40]. Therefore, obesity can be effectively prevented/treated by promoting lipolysis and inhibiting fat formation. Herein, we investigated the anti-obesogenic potential of MSP, BSP, and MBP by evaluating adipogenesis, lipogenesis, and fatty oxidation in 3T3-L1 cells and BPA-induced obese mice. Results showed that MSP, BSP, and MBP are not cytotoxic, and BSP and MBP significantly reduced lipid accumulation in adipocytes. Adipose tissue is mainly composed of large lipid-laden adipocytes, and it is a determinant of energy homeostasis [41,42]. The present study also showed that BSP and MBP administration reduces the adipose tissue weight and body weight in BPA-induced obese mice (*p* < 0.05). Obese individuals usually have higher levels of LDL-c, TG, and TC, as well as lower HDL-c levels, in their blood [43,44]. Our results showed that BSP and MBP significantly lowered the TG levels and increased the levels of HDL-c in the BPA-induced obese mouse model. Obesity is due to excessive or abnormal accumulation of adipose tissue in the body, caused by an increase in the number of lipid accumulation in mature adipocytes [45]. Adipogenesis is a process by which undifferentiated progenitor cells differentiate into mature adipocytes. A complex network of key transcription factors is involved in regulating adipogenesis, and lipogenic factors play important role in controlling lipid metabolism during differentiation [46]. Therefore, obesity can be controlled by regulating adipocyte differentiation at the cellular level. We found that both MBP and BSP treatment suppressed the expression of proteins associated with lipogenesis and adipogenesis (C/EBP-α, PPAR-γ, aP2, SREBP-1C, and FAS) in the BPA-induced 3T3-L1 cells and obese mice model. In addition, we observed that BSP and MBP significantly up-regulated the protein levels of CPT-1 and UCP-1 in adipocytes and epididymal adipose tissue. These results suggest that BSP and MBP treatment reduces fat accumulation and promotes fatty acid oxidation by suppressing lipogenesis and adipogenesis. Thus, BSP and MBP exert anti-obesogenic effects by inhibiting the expression of adipogenesis-related proteins, enhancing lipolysis, and promoting fatty acid oxidation. AMPK is an important energy sensor that regulates energy metabolism in various tissues [47]. AMPK plays a key role in thermogenesis, adipose tissue development, and fatty acid metabolism [48]. Activated AMPK phosphorylates ACC, which inhibits fatty acid synthesis, promotes fatty acid oxidation, and decreases malonyl-CoA levels, thereby restoring CPT-1 activity [49]. AMPK also regulates FAS, SREBP-1C, PPAR-γ, and C/EBP-α expression to affect lipid metabolism and adipogenesis [50,51]. Our results showed that BSP and MBP treatment activates AMPK in BPA-induced 3T3-L1 cells and obese mice. Furthermore, BSP and MBP treatment increased the phosphorylation of ACC, the AMPK substrate, in vitro and in vivo, thus indicating that BSP and MBP can suppress fatty acid synthesis by inhibiting ACC. In summary, our results suggest that BSP and MBP could be effective in the treatment and prevention of BPA-induced obesity.

## 5. Conclusions

BSP and MBP inhibit adipocyte differentiation, reduce lipid accumulation in vitro, and significantly reduce body weight and epididymal adipose tissue mass in BPA-induced obese mice by reducing the expression of adipogenesis-related proteins (aP2, PPAR-γ, SREBP-1C, FAS, and C/EBP-α) and increasing the expression of fatty acid oxidation proteins (CPT-1 and UCP-1). Moreover, BSP and MBP exert anti-obesogenic effects by activating the AMPK signaling pathway. Overall, the alterations in protein expression, associated with the biochemical and physiological improvements induced by BSP and MBP, suggest that BSP and MBP are effective candidates for anti-obesogenic drugs.

## Figures and Tables

**Figure 1 nutrients-14-03814-f001:**
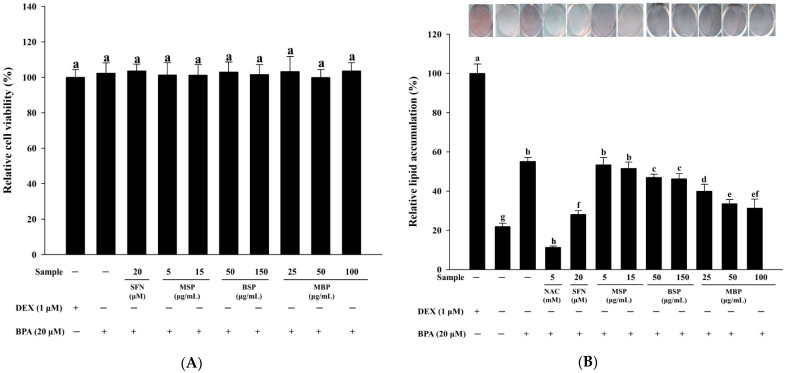
Effects of MBP, BSP, and MSP on cell viability and intracellular lipid accumulation in BPA-induced 3T3-L1 cells. Cell viability by XTT assay (**A**). Lipid accumulations were measured by ORO staining (**B**). Each value is expressed as mean ± SD of three independent experiments. Different lower letters on the bars indicate statistical differences at *p* < 0.05.

**Figure 2 nutrients-14-03814-f002:**
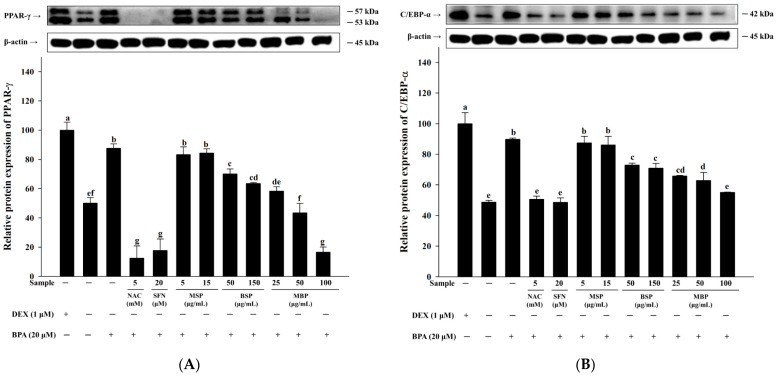
Effects of MBP, BSP, and MSP on the expression of lipid metabolism-related proteins (PPAR-γ (**A**), C/EBP-α (**B**), aP2 (**C**), FAS (**D**), CPT-1 (**E**), and UCP-1 (**F**)) in BPA-induced 3T3-L1 adipocytes were assessed by western blotting. Each value is expressed as mean ± SD of three independent experiments. Different lower letters on the bars indicate statistical differences at *p* < 0.05.

**Figure 3 nutrients-14-03814-f003:**
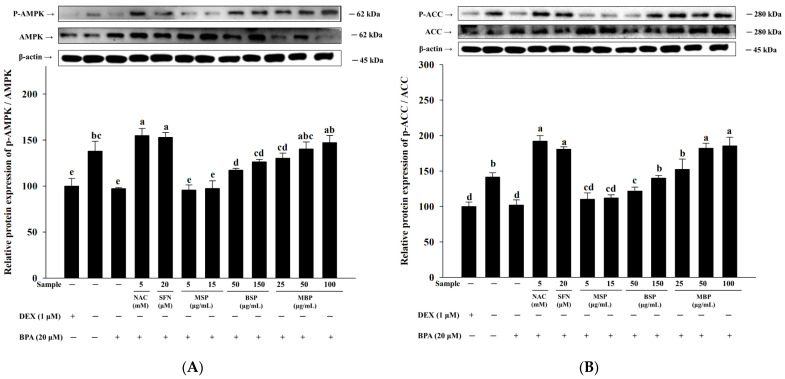
Effects of MBP, BSP, and MSP on the activation of AMPK pathway-related proteins (p-AMPK/AMPK (**A**), p-ACC/ACC (**B**)) in BPA-induced adipocytes. Each value is expressed as mean ± SD of three independent experiments. Different lower letters on the bars indicate statistical differences at *p* < 0.05.

**Figure 4 nutrients-14-03814-f004:**
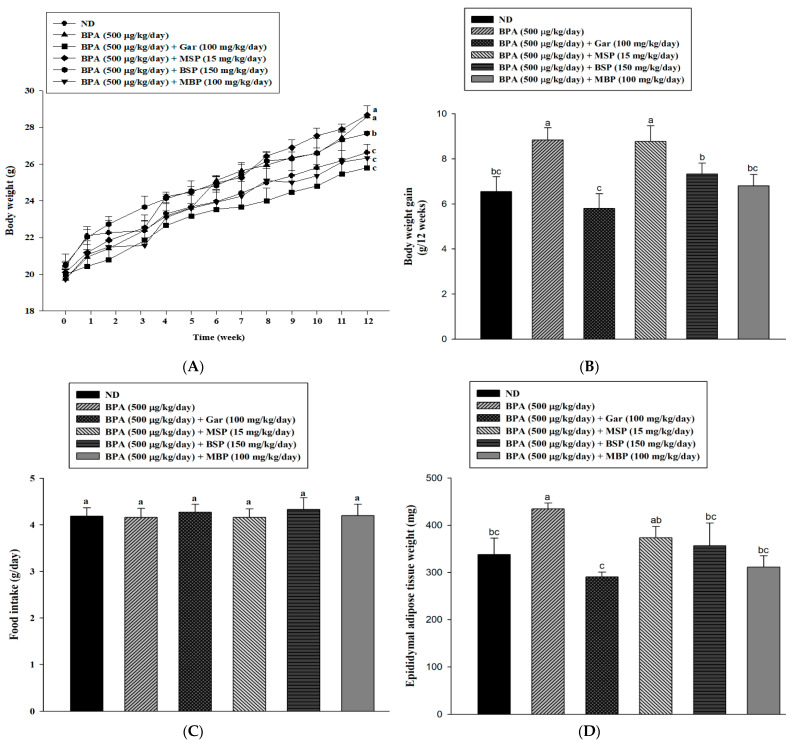
Effects of MBP, BSP, and MSP on adipose tissue structure, tissue weight, and body weight in C57BL/6J mice with BPA-induced obesity. Epididymal adipose tissue sections were stained with H&E and observed under a microscope at 400× magnification. Changes in total body weight (**A**). Body weight gain over 12 weeks (**B**). Food intake (**C**). Epididymal adipose tissue weight (**D**). Other tissue weight (**E**). Histology of the epididymal adipose tissue (**F**). Each value is expressed as mean ± SD (*n* = 5). Different lower letters on the bars indicate statistical differences at *p* < 0.05.

**Figure 5 nutrients-14-03814-f005:**
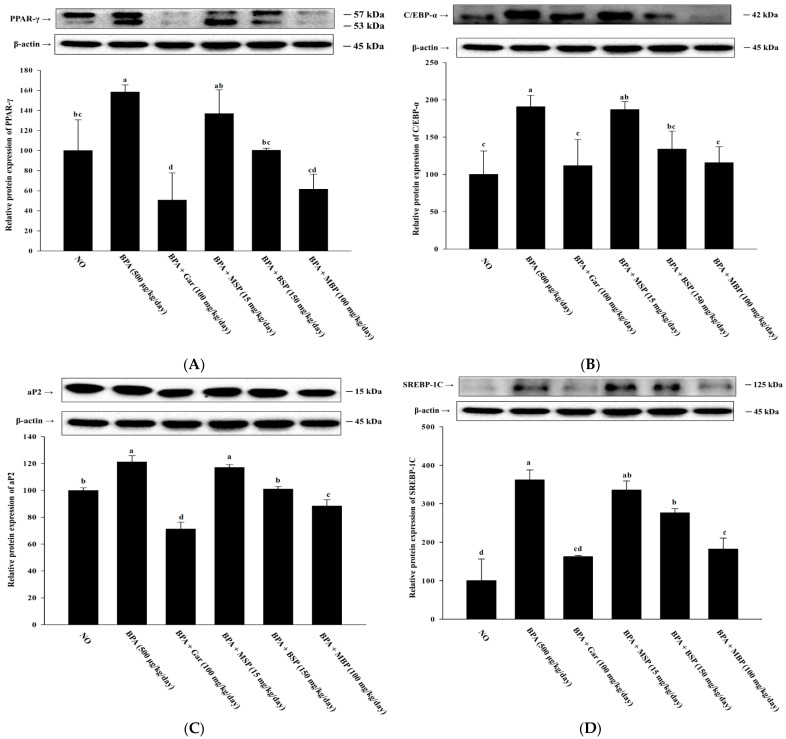
Effects of MBP, BSP, and MSP on the expression of lipid metabolism-related proteins (PPAR-γ (**A**), C/EBP-α (**B**), aP2 (**C**), SREBP-1C (**D**), FAS (**E**), CPT-1 (**F**), UCP-1 (**G**)) in epididymal adipose tissue. Each value is expressed as mean ± SD of three independent experiments. Different lower letters on the bars indicate statistical differences at *p* < 0.05.

**Figure 6 nutrients-14-03814-f006:**
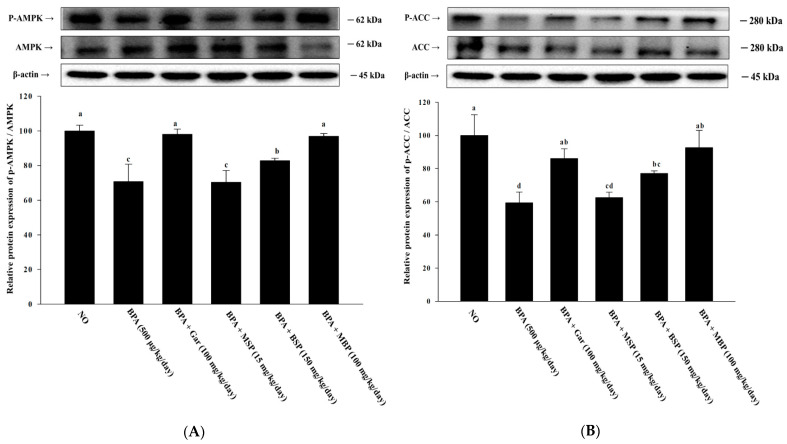
Effects of MBP, BSP, and MSP on the activation of AMPK pathway proteins (p-AMPK/AMPK (**A**), p-ACC/ACC (**B**)) in epididymal adipose tissue. Each value is expressed as mean ± SD of three independent experiments. Different lower letters on the bars indicate statistical differences at *p* < 0.05.

**Table 1 nutrients-14-03814-t001:** Effects of MBP, BSP, and MSP on serum biochemistry in BPA-induced obese C57BL/6J mice.

Groups	ND	BPA (500 μg/kg/day)	BPA + Gar (100 mg/kg/day)	BPA + MSP (15 mg/kg/day)	BPA + BSP (150 mg/kg/day)	BPA + MBP (100 mg/kg/day)
TG (mg/dL)	29.60 ± 6.86 ^ab^	36.42 ± 18.20 ^a^	23.04 ± 2.29 ^ab^	26.17 ± 4.16 ^ab^	19.82 ± 3.24 ^b^	18.84 ± 1.17 ^b^
TC (mg/dL)	60.11 ± 4.41 ^a^	55.53 ± 8.59 ^a^	48.66 ± 2.76 ^a^	58.90 ± 4.44 ^a^	55.06 ± 12.55 ^a^	60.09 ± 6.32 ^a^
HDL-C (mg/dL)	57.95 ± 4.48 ^a^	45.38 ± 2.89 ^b^	56.95 ± 1.14 ^a^	57.03 ± 4.13 ^a^	57.17 ± 0.02 ^a^	60.09 ± 5.37 ^a^
LDL-C (mg/dL)	9.53 ± 0.64 ^a^	9.82 ± 5.20 ^a^	8.98 ± 0.93 ^a^	10.47 ± 0.51 ^a^	10.73 ± 0.66 ^a^	10.31 ± 2.33 ^a^

Data are expressed as mean ± SD (*n* = 5). Superscript letters represent significant differences with *p* <0.05.

## Data Availability

The data presented in this study are available within this article.

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
