# Peer review of "Anti-Obesogenic Effects of Sulforaphane-Rich Broccoli (Brassica oleracea var. italica) Sprouts and Myrosinase-Rich Mustard (Sinapis alba L.) Seeds In Vitro and In Vivo"

_nutrients, 2022, doi:10.3390/nu14183814_

Round 1
Reviewer 1 Report
The issue of the paper is interesting. The methods used for the effectiveness of compounds are solidly presented. Isolation and detection of compounds are adequate. The action of compounds on obesity is adequately done. All relevant indicators are shown and well explained. Maybe it wouldn't be bad to check it again on rats, and then move on to the clinical application.
Author Response
We appreciate you for the time and effort on our paper.
Reviewer 2 Report
The authors have carried out a good experimental work, contributing with their results to the field of the use of nutraceuticals in the treatment of obesity. This original article is well organized and contains the necessary elements however needs some minor revisions.
1. Update references 1 and 2.
2. Line 410 put the word in vivo in italics.
3. Improve the size and quality of the text of all the figures, many are too small and others are blurred.
Author Response
Dear reviewer,
We appreciate you for the time and effort on our paper. The responses to reviewer comments have been marked in red throughout the manuscript. We hope the changes have addressed all the shortcomings outlined. Below you could find our point-by-point response to reviewer comments.
Response 1: Update references 1 and 2.
AU: Thanks for your suggestions very much. We have updated references “1. Qi, L.; Cho, Y.A. Gene-environment interaction and obesity. Nutr. Rev. 2008, 66, 684-694." and "2. Jo, J.; Gavrilova, O.; Pack, S.; Jou, W.; Mullen, S.; Sumner, A.E.; Cushman, S.W.; Periwal, V. Hypertrophy and/or hyperplasia: dynamics of adipose tissue growth. PLoS Comput. Biol. 2009, 5, e1000324.” to “1. Jia, P. Obesogenic environment and childhood obesity. Obes. Rev. 2020, 22, e13158.” And “2. Liu, F.; He, J.; Wang, H.; Zhu, D. ; Bi, Y. Adipose morphology: a critical factor in regulation of human metabolic diseases and adipose tissue dysfunction. Obes. Surg. 2020, 30, 5086–5100.” Please see lines 407-409.
Response 2. Line 410 put the word in vivo in italics.
AU: Thanks for your suggestions very much. We have italicized the word "in vivo". Please see line 376.
Response 3. Improve the size and quality of the text of all the figures, many are too small and others are blurred.
AU: Thanks for your suggestions very much. We have revised all figures in the manualscript and improved the quality, clarity and text size of the figures. Please see the figure section on pages 5-13.
